# Atomic Model of Gold Adsorption onto the Pyrite Surface with DFT Study

Chunlin Liu [1], Yongbing Li [2,*], Qi Cheng [2] and Yang Zhao [2]

[1]   College of Earth and Planetary Science, University of Chinese Academy of Sciences, Beijing 100049, China; liuchunlin19@mails.ucas.ac.cn
[2]   Key Laboratory of Computational Geodynamics, University of Chinese Academy of Sciences, Beijing 100049, China; chengqi171@mails.ucas.ac.cn (Q.C.); zhaoyang18@mails.ucas.edu.cn (Y.Z.)
*    Correspondence: yongbingli@ucas.ac.cn

**Abstract:** Adsorption mineralization of gold is an important mineralization mechanism under epigenetic and low temperature conditions. In this paper, a plane-wave pseudopotential method based on density functional theory (DFT) is used to explore the adsorption mechanism of gold on the surface of pyrite. Among the three surfaces of pyrite, the surface energies of (100), (111), and (210) surfaces are 1.0508, 1.5337, and 1.8255 J·m$^2$, respectively, and the (100) surface is the most stable surface in the thermodynamic state. The adsorption capacities of gold atoms under different surfaces are (210) ($-2.68$ eV) > (111) ($-1.67$ eV) > (100) ($-0.84$ eV). Mulliken analysis indicates that charge transfer occurs after the adsorption of gold atoms onto the surface of pyrite (210), and gold and iron atoms are oxidized with the reduction of sulfur atoms. The density of states (PDOS) analysis shows that the 5d orbital on the Fermi energy level of the iron atom is active and the adsorption capacity is greater than that of the sulfur atom, and adsorption is formed between the gold atom, which leads to the gold being able to be stably deposited on the surface of pyrite (210).

**Keywords:** pyrite; Au; adsorption; density functional theory (DFT)

## 1. Introduction

Pyrite is a common gold-bearing mineral, and the surface properties of pyrite play an important role in the adsorption of gold [1,2]. Previous studies on the adsorption of free gold ions by pyrite and the mechanism of enrichment mineralization mainly focus on the geological conditions of mineralization [3], mineral element characteristics [4,5], fluid activity [6,7], etc., while the microscopic mechanism of gold ion adsorption on the surface of pyrite is relatively limited. Current studies on the genesis of gold-bearing pyrite deposits mainly focus on macroscopic mechanisms, often from the perspective of isotopic geochemistry [8,9], tectonic activity [10], and fluid action [11]. The study of these macroscopic mechanisms has played an important role in the exploration of gold-bearing pyrite.

There are two main mechanisms for controlling the precipitation of gold on mineral surfaces [12,13]: solubility control mechanism and mineral surface action mechanism. Among them, the mineral surface action mechanism of gold, that is, gold-bearing hydrothermal fluids (especially gold-unsaturated hydrothermal fluids) in which gold complexes are reduced by adsorption on the surface of sulfide minerals and lead to gold precipitation [1,14]. The adsorption mineralization of gold makes the precipitation enrichment of gold is no longer limited by the solubility of gold, but controlled by the solid–liquid interface reaction between sulfide minerals and gold complex solution [15–17], especially in epigenetic and low temperature conditions, adsorption is an important mineralization mechanism, so it is important to study the microscopic mechanism of gold adsorption on the surface of pyrite.

First principle calculation is based on Schrödinger's equation [18,19], making the necessary assumptions about the actual system and the problem under study, and then performing numerical simulations. It starts from the atomic components and electronic structure of the substance to be studied, applies quantum mechanics and other physical laws, and determines the geometry, electronic structure, thermodynamic properties, optical properties, and other physical properties of the specified substance through self-consistent calculations.

In recent years, more and more results have been obtained on the surface properties of pyrite using the first principle, and most of the studies have focused on the study of the physical properties of pyrite. Blanchard calculated arsenic incorporation into pyrite by density functional theory (DFT) and the effect of subsequent incorporation on the dissolution of pyrite crystals [20]; Liu et al. calculated the thermodynamic properties of pyrite at high temperature and pressure using a quasi-harmonic approximation using a first-principles approach based on density flooding perturbation theory [21].

The crystals of pyrite belong to the cubic crystal system. Pyrite in nature has different morphologies and specific dissociation surfaces, such as cubic, octahedral, and pentadodecahedral. Hung studied the properties of different surfaces of pyrite [22,23] and determined the thermodynamic stability of (100), (110), (111), and (210) surfaces, respectively, with different thickness choices, cutoff energies, K-point lattices in the Brillouin zone, and stochastic and non-random models. A preliminary interpretation of the observed crystal morphology is given based on the calculation of the interatomic interaction potential. The vacuum contains the growth of all surfaces, with a dominant growth of (100) surface cubes and a strong stability of pentadodecahedral growth with octahedral growth on (210) and (111) surfaces. The growth of different surfaces depends on the influence of factors such as partial pressure and temperature during mineralization. Meng [24] simulated the morphological evolution of pyrite under gold precipitation conditions. By X-ray powder diffraction and transmission electron microscopy analysis, it was concluded that the morphology of pyrite changed from a combination of (210) and (111) to a more stable combination of (111) and (100) or a single morphology when the molar ratio was different and the temperature was varied.

Most previous studies have been conducted from a macroscopic macromolecular perspective, which cannot be refined to the atomic scale, and thus cannot fully characterize the microscopic mechanism of adsorption. The microscopic mechanism of adsorption is studied to determine the microscopic mechanism of adsorption occurrence by accurately grasping the difference in the surface electron distribution between the surface atoms of minerals and adsorbed atoms before and after adsorption. In this paper, the microscopic adsorption mechanism of gold particles on the surface of pyrite is investigated using the first nature principle. The magnitudes of the three surface energies of (100), (111) and (210) were calculated successively to elucidate one of the causes of the presence of different surfaces of pyrite crystals. In addition, by measuring the difference of electron distribution before and after adsorption, the adsorption energy magnitude of gold on the three surfaces of py-rite was compared, as well as the interatomic interactions during the adsorption process. The ability of adsorption of gold atoms on the (100), (111), and (210) surfaces is explained from the microscopic mechanism. The in-depth study of adsorption properties on the surface of pyrite crystals explores the causes of the differences in floatability of pyrite in different deposits from a new perspective, which has certain theoretical and practical application reference values for improving the flotation technology of pyrite and enriching the basic theory of flotation.

## 2. Calculational Methods and Models

### 2.1. Computational Methods

The calculations are based on density functional theory (DFT) [25,26], using the local density approximation (LDA) [27] and the generalized gradient approximation (GGA) [28] for the exchange association energy. The Revised Perdew–Burke–Ernzerhof functional

(RPBE) [29] generalized function has high energy and structure prediction in accuracy and is used to model electron exchange and association interactions in GGA.

The geometric optimization of the initial pyrite cell was performed using the CASTEP module under the Materials Studio software based on first principles [30,31], and different exchange generalized functions and plane wave truncation energies were selected to calculate the simulated optimization data. The interaction between electrons and ions is described by the UltraSoft pseudopotential (USPP) [32,33]. The valence electrons involved in the calculation of the pseudopotential are S $3s^23p^4$; Fe $3p^63d^64s^2$; Au $5p^65d^{10}6s^1$. The atomic positions were optimized by using the Broyden–Fletcher–Goldfarb–Shanno (BFGS) [34] algorithm. The Monkhorst–Pack [35] method was used for the Brillouin zone integral. The positions of all atoms in the reciprocal space in the calculation are relaxed according to Hartree–Fock [36].

The calculations of atomic density of states, energy band diagrams, and Mulliken bonding residences within the mineral protocells are then based on the geometrically optimized model. During the calculation, the self-consistent field (SCF) [37] convergence accuracy is set to $5 \times 10^{-7}$ eV/atom, and the pseudopotential is chosen as the super soft pseudopotential. The convergence parameters for geometric optimization are set: the convergence threshold of the interatomic interaction force is 0.1 eV/nm, the internal crystal stress is not greater than 0.02 GPa, the maximum atomic displacement is not greater than $5 \times 10^{-3}$ nm, and the number of iterations does not exceed 100. All calculations are performed in the reciprocal space.

To determine the optimal parameters for the optimization of the pyrite cell structure by the CASTEP module, different DFT methods were first selected for the calculations. Other parameters were chosen as the default k-point sampling $4 \times 4 \times 4$ with cut off energy of 340 eV. The comparison of the simulation results for the optimization of pyrite crystals by choosing different exchange correlation function with the experimental results of Wu' experimental results [38] was shown in Table 1.

**Table 1.** Exchange–Correlation Functions convergence test results for cubic pyrite.

| Exchange–Correlation Functions/eV | Lattice Parameter/Å | Parameter Deviation Δc/% | Total Energy/eV |
|---|---|---|---|
| LDA-CA-PZ | 5.2566 | 2.9270 | −5696.3077 |
| GGA-PBE | 5.3771 | 0.7017 | −5692.2264 |
| GGA-RPBE | 5.4183 | 0.0591 | −5696.2807 |
| GGA-PW91 | 5.3813 | 0.6242 | −5702.4578 |
| GGA-WC | 5.3209 | 1.7396 | −5683.4191 |
| GGA-PBESOL | 5.3086 | 1.9667 | −5671.2324 |
| Experimental results [38] | 5.4151 | | |

According to Table 1, it can be seen that the lattice parameters obtained by the calculation using the GGA-RPBE method have the smallest deviation from the experimental data, and the deviation is less than 1%. The GGA-RPBE exchange correlation function was selected for the next calculation. Different k-point sampling and cut off energy parameters were set to calculate the energy and cell parameters of the pyrite cell system, as shown in Table 2.

From the calculation results in Table 2, it can be seen that, when the k-point sampling is set at $3 \times 4 \times 4$ and the cut off energy was at 350 eV, the deviation was the smallest compared with the experimental data, and the system energy was basically stable. Considering the reliability and computational efficiency of the simulation, the k-point sampling of the simulation calculation was therefore chosen as $3 \times 4 \times 4$, and the cut off energy was chosen as 350 eV.

**Table 2.** Results of cut off energy convergence and k point tests of cubic pyrite.

| | Calculation Setting | Lattice Parameter/Å | Parameter Deviation Δc/% | Total Energy/eV |
|---|---|---|---|---|
| | 310.0000 | 5.4602 | 0.8329 | −5696.3292 |
| | 330.0000 | 5.4214 | 0.1163 | −5696.2412 |
| Cut off Energy/eV | 350.0000 | 5.4162 | 0.0203 | −5696.2679 |
| | 370.0000 | 5.4084 | 0.1237 | −5696.2967 |
| | 390.0000 | 5.4080 | 0.1311 | −5696.3132 |
| | $3 \times 3 \times 3$ | 5.4170 | 0.0351 | −5696.3030 |
| k Point | $3 \times 4 \times 4$ | 5.4167 | 0.0295 | −5696.3053 |
| | $3 \times 3 \times 4$ | 5.4168 | 0.0314 | −5696.3023 |
| Experimental results [38] | | 5.4151 | | |

*2.2. Effect of Surface Energy*

Pyrite crystals belong to the isometric crystal system, each cell unit contains 4 $FeS_2$ molecular units, iron atoms are distributed on six face centers and eight top angles of cubic cells, and each iron atom is coordinated with six adjacent sulfur atoms to form an octahedral structure, while each sulfur atom is coordinated with three iron atoms and one sulfur atom to form a tetrahedral structure, and two sulfur atoms form a dumbbell-like structure between them, in the form of sulfur dimer ($S_2^{2-}$). According to the thermodynamic minimum Gibbs free energy principle, the final equilibrium state of the crystal should be the state when the surface Gibbs free energy is at its lowest, and the analysis of the crystalline surface Gibbs free energy of pyrite can determine the best adsorption surface.

After optimizing the geometry of pyrite crystals, three different surface models were cleaved in the (100) crystal direction, (111) crystal, and (210) crystal direction, respectively, as shown in Figure 1.

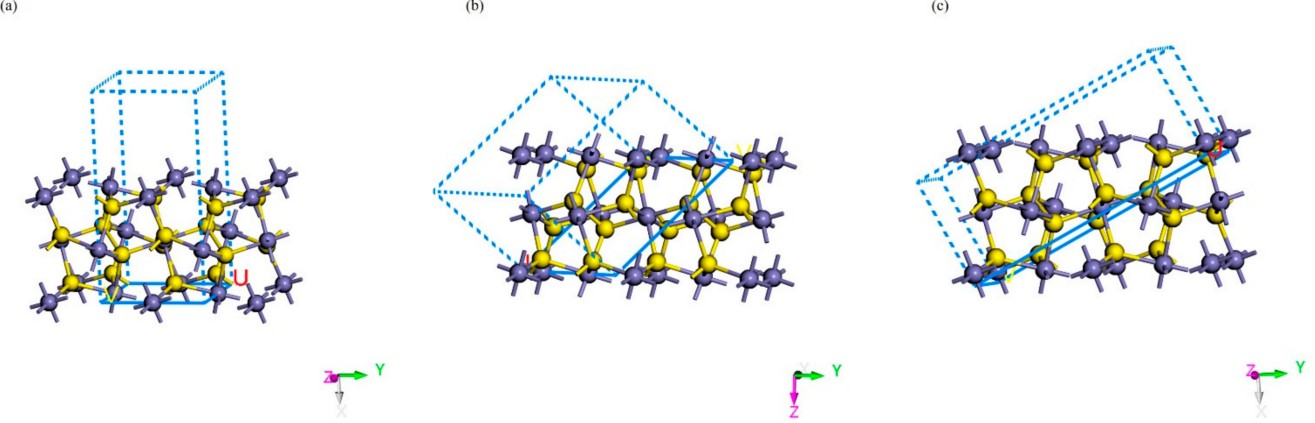

**Figure 1.** Pyrite crystal cleavage sites: (**a**) (100) Surface cleaved sites; (**b**) (111) Surface cleaved sites; (**c**) (210) Surface cleaved sites. (The yellow ball: S; the blue ball: Fe).

The surface energy level of the model in density functional theory is mainly affected by the atomic layers, thickness of the vacuum layer, and surface relaxation [39]. Therefore, the surface energy was calculated for different surface atomic layers and thickness of the vacuum layer for (100), (111) and (210), respectively. The calculated results in Table 3 show the optimal atomic layer thickness and vacuum layer thickness for the three surfaces with the lowest surface energy as a choice of parameters for further calculations.

**Table 3.** Surface energy of different surfaces of pyrite.

| Surface | Atomic Layers | Thickness of the Vacuum Layer/Å | Surface Energy/J·m$^2$ |
|---------|---------------|---------------------------------|------------------------|
| (1 0 0) | 18 | 10 | 1.0508 |
| (1 1 1) | 15 | 10 | 1.5337 |
| (2 1 0) | 15 | 10 | 1.8255 |

*2.3. Adsorption Forms of Gold on the Surface of Pyrite*

Metallic gold is deposited from chloride solutions containing $Au^{3+}$ in two steps: Reduction of $Au^{3+}$ to $Au^+$, which is rate-determining, followed by a much faster reduction of $Au^+$ to $Au^0$ [40]. The deposition process is mainly controlled by the reduction rate of $Au^{3+}$; the oxidation rate of the pyrite surface is relatively slow. In addition, the deposition process is influenced by the Fe(II)/Fe(III) redox reaction on the mineral surface, independent of the reduction of gold complexes and the oxidative coupling of metal sulfides, and the adsorption of gold increases with the decrease of the electrochemical potential of the mineral.

The gold adsorbed on the surface of pyrite is 0-valent gold nanoparticles. Yuri [2] experimentally investigated the gold species spontaneously deposited on pyrite in $HAuCl_4$ solution at room temperature and the process of its reaction mineral surface. Gold was found to precipitate as gold atoms and tend to accumulate to form larger particles. SEM photographs of the pyrite surface showed that the size and number of gold particles increased with time during the adsorption of gold on the pyrite surface in $Au^{3+}$ chloride solution.

The DFT method is used to help understand the adsorption mechanism of gold particles by constructing a model for the adsorption of individual gold atoms on different surfaces of pyrite, so this study focuses on the calculation of 0-valent gold on the surface of pyrite.

**3. Results and Discussion**

After adding the gold atoms to different surfaces, the models of gold atoms with pyrite (100), (111), and (210) systems after sufficient relaxation by geometric optimization are shown in Figure 2. It can be seen that there are differences in the positions of gold atoms adsorbed on different surfaces. On the (100) surface, gold atoms are attracted by three sulfur atoms and one iron atom to reach a stable structure, and the (111) and (210) surfaces are attracted by two sulfur atoms and two iron atoms, respectively, to form a stable structure. In addition, for different surfaces, gold atoms and neighboring atoms between the distances are different, (210) surface gold atoms and neighboring atoms between the distances are smaller than the (100) surface, and (111) surface gold atoms and neighboring atoms between the distances may be because the pyrite (210) surface gold atoms adsorption effect is stronger than (100) surface and (111) surface. Likewise, the charge distribution of surface atoms differs before and after adsorption of mineral crystals, and, by analyzing the change of charge values, it can be used to further reveal the adsorption mechanism.

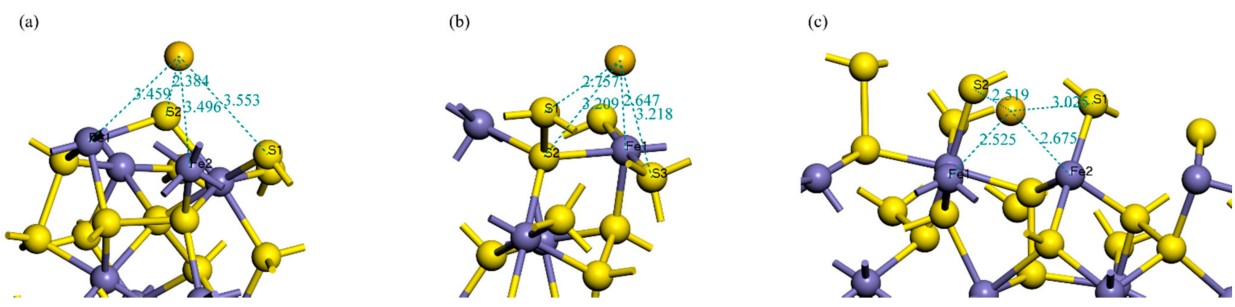

**Figure 2.** Geometry optimization model after adsorption of gold atoms on pyrite (**a**) (100), (**b**) (111), and (**c**) (210) surface. (The yellow ball: S; the blue ball: Fe; the gold ball: gold).

### 3.1. Adsorption Energy and Stable Adsorption Configuration

The adsorption energy can be used to reflect the stability of the adsorbed material on the adsorption surface, and the adsorption energy is calculated as Equation (1):

$$E_{adsorption} = E_{adsorbate/slab} - E_{adsorbate} - E_{slab} \tag{1}$$

in which $E_{slab}$ is the adsorption energy on the surface of pyrite, $E_{adsorbate/slab}$ is the total adsorption energy of the system after adsorption occurs, and C and D are the energy of gold atoms and the surface energy of pyrite before adsorption. When the adsorption energy is negative, the larger the absolute value of the negative value, the more stable the adsorption is. Therefore, the optimization of adsorption conformation is to find the adsorption conformation with the lowest adsorption energy.

The adsorption energies of the system after relaxation on different surfaces were calculated separately. When the gold atom was adsorbed on the pyrite surface, the adsorption energy can be negative, indicating that the system is more stable. In the present work, the adsorption energies were negative after adsorption occurred on different surfaces. The adsorption energies of (210), (111), and (100) surfaces are (−2.68 eV), (−1.67 eV), and (−0.84 eV), respectively. By comparing the magnitudes of the adsorption energies, it is found that the (210) surface has the most stable structure after adsorption of gold atoms, which corresponds to the shortest distance between the adjacent atoms of gold on the (210) surface, and the (210) surface has the strongest ability to adsorb gold atoms, while the (210) surface has the strongest ability to adsorb gold atoms on the (100), and (111) surface adsorption is relatively weak.

### 3.2. Surface Atoms Mulliken Analysis

Mulliken analysis is a method proposed by Mulliken to represent the distribution of charge among constituent atoms, which can examine the charge distribution, transfer and bonding properties of the simulated system [31]. In order to further understand the types of interatomic charge changes into bonds, Mulliken charge analysis was calculated for different surface adsorption systems.

From the calculation results, the surface charge of the initial adsorbate individual gold atoms before adsorption occurred was 0. After geometric optimization, as shown in Figure 2, the surface charges of the sulfur and iron atoms adjacent to the gold atoms on the (100) surface were 0.06, 0.07, 0.10, and −0.24, corresponding to atomic spacings of 2.757 Å, 3.209 Å, 3.218 Å, and 2.647 Å, respectively. The atomic charges of the surface layer changed considerably and a charge transfer occurred. The sulfur atoms mainly lose electrons, while the iron and gold atoms mainly gain electrons. The population analysis of the electron in different orbitals shows that the sulfur atom loses electrons mainly provided by the p orbital, and the s orbital loses a small number of electrons, in which the sulfur atom with the closest distance to the gold atom contributes the most electrons, losing 0. 14 electrons, and the remaining two sulfur atoms lose 0.05 and 0.08 electrons, respectively; the iron atom gains a small number of electrons in the s orbital, and the most electrons in the p orbital, gaining a total of 0.24 electrons. The Au atom gains some electrons in the s orbital, gains some electrons in the p orbital, and loses some electrons in the d orbital, gaining a total of 0.08 electrons, with a surface charge of −0.08 e.

After the adsorption on the surface of (111), the surface charges of the sulfur and iron atoms adjacent to the gold atom are 0.66, 0.66, −1.42, and −1.51, respectively, and the corresponding atomic distances are 3.553 Å, 2.384 Å, 3.459 Å, and 3.496 Å. Similarly, in the process of adsorption, the sulfur atoms mainly lose electrons, iron atoms gain electrons, the more special is the gold atom loses a small number of electrons. The electrons lost by the sulfur atom in the (111) plane is mainly provided by the s orbital, and the p orbital is almost unchanged; in addition, the iron atom in the (111) plane gains the most electrons in the p orbital, and gains a few electrons in both the s and d orbitals, with a large change in

surface charge; the Au atom gains more electrons in the p orbital and loses some electrons in the s and d orbitals, with a total loss of 0.03 electrons, and the surface charge is 0.03 e.

On the (210) surface, the surface charges of the sulfur and iron atoms adjacent to the gold atoms are 0.41, 0.28, −0.63, and −0.68, respectively, corresponding to atomic spacings of 3.025 Å, 2.519 Å, 2.525 Å, and 2.675 Å. According to the results of the Mulliken population analysis, the sulfur atoms lose charge and the iron and gold atoms gain charge. The charge lost by the sulfur atoms on the (210) surface is also mainly provided by the s orbital, and the p orbital electrons change less; in addition, the charge gained by the iron atoms is concentrated in the p orbital, and a small amount of charge is gained in the s orbital, and the surface charge changes more; the Au atoms gain more charge in the s orbital and lose a large amount of charge in the d orbital, and the total charge is −0.11 e.

From the atomic Mulliken charge analysis, gold atoms in the adsorption process on the surface of pyrite can be seen, S atoms to the surface of the metal atoms transferred part of the charge, gold atom surface charge changes, and adsorption occurs, as shown in Table 4.

**Table 4.** Mulliken charge population of atoms before and after Au adsorption on the pyrite surface.

| Surface | Atom | | s | p | d | Total | Charge/e |
|---|---|---|---|---|---|---|---|
| (100) | S1 | Before | 1.85 | 4.23 | 0.00 | 6.08 | −0.08 |
| | | After | 1.84 | 4.10 | 0.00 | 5.94 | 0.06 |
| | S2 | Before | 1.81 | 4.16 | 0.00 | 5.98 | 0.02 |
| | | After | 1.81 | 4.13 | 0.00 | 5.93 | 0.07 |
| | S3 | Before | 1.81 | 4.16 | 0.00 | 5.98 | 0.02 |
| | | After | 1.80 | 4.10 | 0.00 | 5.90 | 0.10 |
| | Fe1 | Before | 0.36 | 0.48 | 7.15 | 7.99 | 0.01 |
| | | After | 0.37 | 0.79 | 7.09 | 8.24 | −0.24 |
| | Au | Before | 1.00 | 0.00 | 10.00 | 11.00 | 0.00 |
| | | After | 1.19 | 0.09 | 9.81 | 11.08 | −0.08 |
| (111) | S1 | Before | 1.89 | 4.31 | 0.00 | 6.20 | −0.20 |
| | | After | 1.04 | 4.30 | 0.00 | 5.34 | 0.66 |
| | S2 | Before | 1.89 | 4.31 | 0.00 | 6.20 | −0.20 |
| | | After | 1.01 | 4.33 | 0.00 | 5.34 | 0.66 |
| | Fe1 | Before | 0.38 | 0.48 | 6.98 | 7.84 | 0.16 |
| | | After | 0.49 | 1.85 | 7.08 | 9.42 | −1.42 |
| | Fe2 | Before | 0.38 | 0.48 | 6.98 | 7.84 | 0.16 |
| | | After | 0.50 | 1.93 | 7.08 | 9.51 | −1.51 |
| | Au | Before | 1.00 | 0.00 | 10.00 | 11.00 | 0.00 |
| | | After | 0.98 | 0.18 | 9.81 | 10.97 | 0.03 |
| (210) | S1 | Before | 1.86 | 4.26 | 0.00 | 6.12 | −0.12 |
| | | After | 1.40 | 4.19 | 0.00 | 5.59 | 0.41 |
| | S2 | Before | 1.86 | 4.40 | 0.00 | 6.26 | −0.26 |
| | | After | 1.41 | 4.31 | 0.00 | 5.72 | 0.28 |
| | Fe1 | Before | 0.39 | 0.34 | 7.06 | 7.79 | 0.21 |
| | | After | 0.49 | 1.09 | 7.05 | 8.63 | −0.63 |
| | Fe2 | Before | 0.35 | 0.36 | 7.09 | 7.80 | 0.20 |
| | | After | 0.46 | 1.14 | 7.08 | 8.68 | −0.68 |
| | Au | Before | 1.00 | 0.00 | 10.00 | 11.00 | 0.00 |
| | | After | 1.25 | 0.15 | 9.70 | 11.10 | −0.11 |

Mulliken bonding analysis can reflect the overlap between atoms, and Mulliken [19] believes that the total overlap charge $n$ (A, B) between bonding atoms A and B can reflect the degree of covalent bond formation between A and B atoms. If $n$ (A, B) is positive, A

and B atoms are bonded, if $n$ (A, B) is large, covalent bonds exist between A and B atoms; if $n$ (A, B) is small, it means that A and B atoms are ionically bonded; if $n$ (A, B) is negative, anti-bonding orbitals are produced, which means that A and B atoms are not bonded.

In order to further study the bonding properties between the atoms on the surface of pyrite and gold atoms, the bonding population of gold atom adsorbed on different surfaces of pyrite was compared, as shown in Table 5. The Au atoms on the (100) surface mainly bond with the adjacent S1 and Fe atoms, with Mulliken population values of 0.03 and 0.36, respectively, and the bonding between Fe and Au atoms is more covalent, the bonding between Au and S1 is ionic, and the adsorption of Fe atoms is stronger than that of S1; (111) surface has only S2 and gold atom into bonding, the population value is 0.41, covalent stronger; (210) surface and gold atom into bonding the most surface atoms, iron atoms and gold atoms bonding population for 0.56 and 0.22, S2 and gold atom into the bonding population for 0.18, and the bonding ability is not as strong as iron atoms. Combined with the comparison of the adsorption energy of different surfaces, it can be seen that the gold atoms in the (210) surface is the most easily adsorbed, the most bonded atoms, and the larger population value; (111) surface, although only a single sulfur atom and gold atom bonded, but the population value is higher, and the atomic spacing is the shortest, the adsorption capacity is second; (100) surface S1 and iron atoms have adsorption effect on gold atoms, adsorption capacity is limited.

**Table 5.** Mulliken bond population of atoms before and after Au adsorption on pyrite (100) surface.

| Surface | Bond | Population | Bond Length/Å |
|---------|------|------------|---------------|
| (100) | Fe–Au | 0.36 | 2.6472 |
|  | S1–Au | 0.03 | 2.7567 |
| (111) | S2–Au | 0.41 | 2.3845 |
| (210) | S2–Au | 0.18 | 2.5191 |
|  | Fe1–Au | 0.56 | 2.5255 |
|  | Fe2–Au | 0.22 | 2.6751 |

*3.3. Band Structure of Au Pyrite (210) Surface Adsorption*

Based on the surface energy band structure calculations, Figure 3 shows the energy band structure changes before and after adsorption of gold atoms on the surface of pyrite (210). The bandgap before and after adsorption are 0.44 and 0.094 eV, respectively, setting the energy zero point at the Fermi energy level (Ep), and the distinct surface states of pyrite (210) can be observed near the Fermi energy level. The gold atom makes the density of states of the pyrite surface layer through the Fermi energy level, and the band gap decreases from 0.44 eV at the ideal surface to 0.094 eV, which improves the conductivity of the pyrite (210) surface. This indicates that the adsorption of gold atoms elevates the electron concentration on the surface of pyrite (210), and the surface conductivity is enhanced.

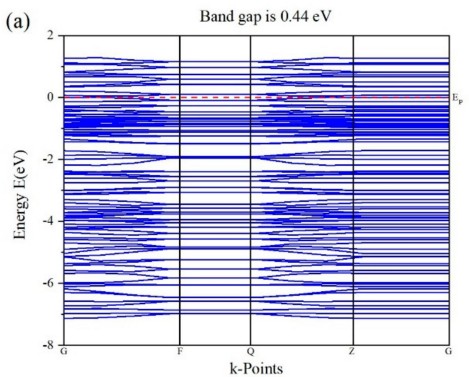 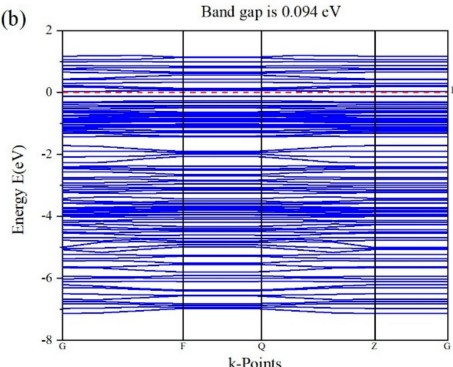

**Figure 3.** Band structure of pyrite (**a**) before Au adsorption, (**b**) after Au adsorption.

### 3.4. Projected Density of States (PDOS) Analysis of Au Pyrite (210) Surface Adsorption

Density of state (DOS) is one of the most important parameters to describe the state of electron motion in solid state physics, and the projected density of state (PDOS) can reflect the state of electron motion in different orbital energy levels separately [22]. In order to gain more insight into the electronic structure changes of surface atoms before and after adsorption, the projection of the fractional wave density of states PDOS was carried out for the Au/$FeS_2$ (210) system, which has the strongest adsorption capacity.

To further investigate the surface effect on pyrite surface, depth-resolved electronic structures of pyrite surface are shown in Figure 4, with the valence electron configuration of Au 5d106s1 for the gold atom. The PDOS of Au atom was composed of 6s, 5p, and 5d orbitals, and the PDOS of Fermi level was mainly contributed by the Au 6s orbital. Before and after adsorption, the d-orbital peak decreases significantly, the s-orbital moves to lower energy levels, and the atomic activity decreases to the steady state. Because the closer to the Fermi energy level, the stronger the electronic activity is, and important physicochemical reactions always occur near the Fermi energy level, the s and d orbitals are involved in the main adsorption reactions as the adsorption energy is enhanced. It can be seen from the density of states plot that the atomic density of states on the surface of pyrite crosses the Fermi energy level before and after adsorption, so the surface of (210) has semiconductor characteristics and exhibits some metallic properties, which is consistent with the calculation results of Hung et al. [23]. In addition, after adsorption, the overall density of states of the (210) surface has limited change, and the density of states of the d orbitals in the valence band part of the left side of the Fermi energy level increases slightly. At the low energy level, the peak of the s orbitals decreases and the peak of the d orbitals increases, which may be caused by the interaction of the gold atoms with the surface atoms.

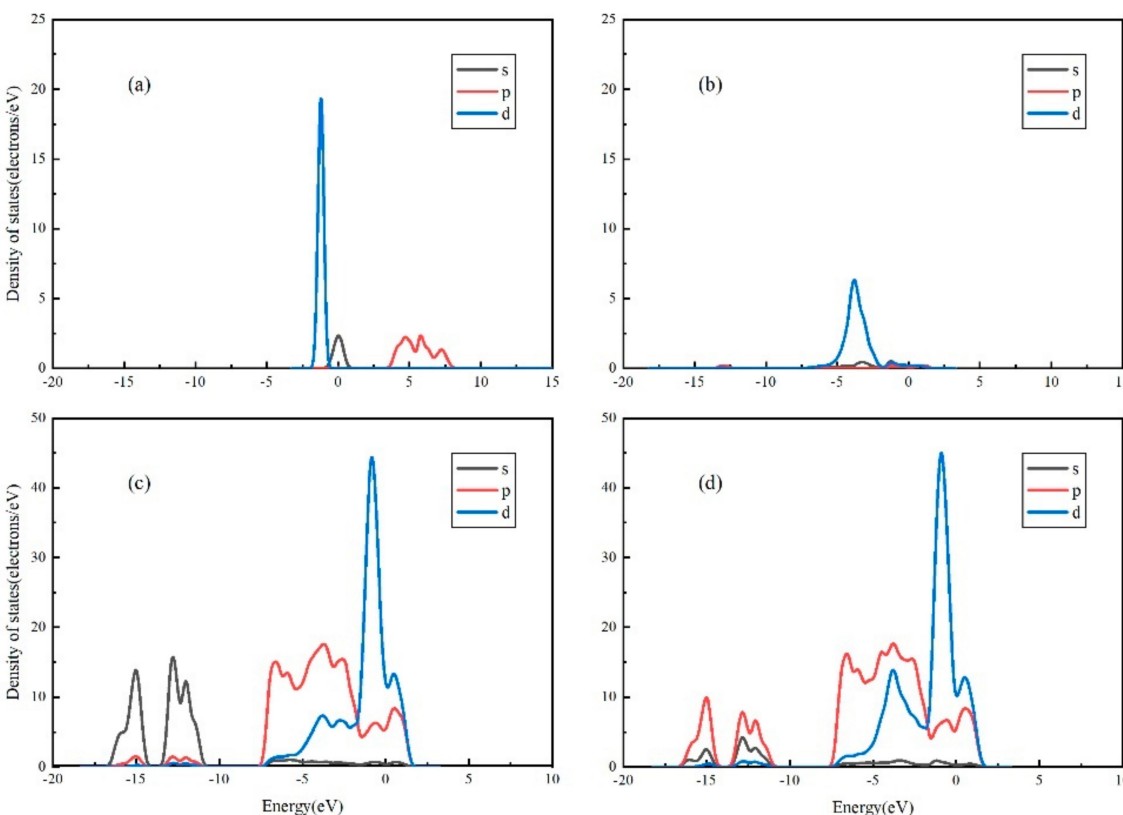

**Figure 4.** The PDOS plots for the Au atom and the pyrite surface atoms: (**a**) for Au before adsorption; (**b**) for Au adsorption on pyrite (210) surface; (**c**) for pyrite (210) surface before adsorption; (**d**) for pyrite (210) surface after Au is adsorbed.

To further illustrate the activation effect of gold atoms adsorbed on pyrite surface atoms, PDOS was analyzed separately for S2, Fe1 and Fe2 atoms, as shown in Figure 5.

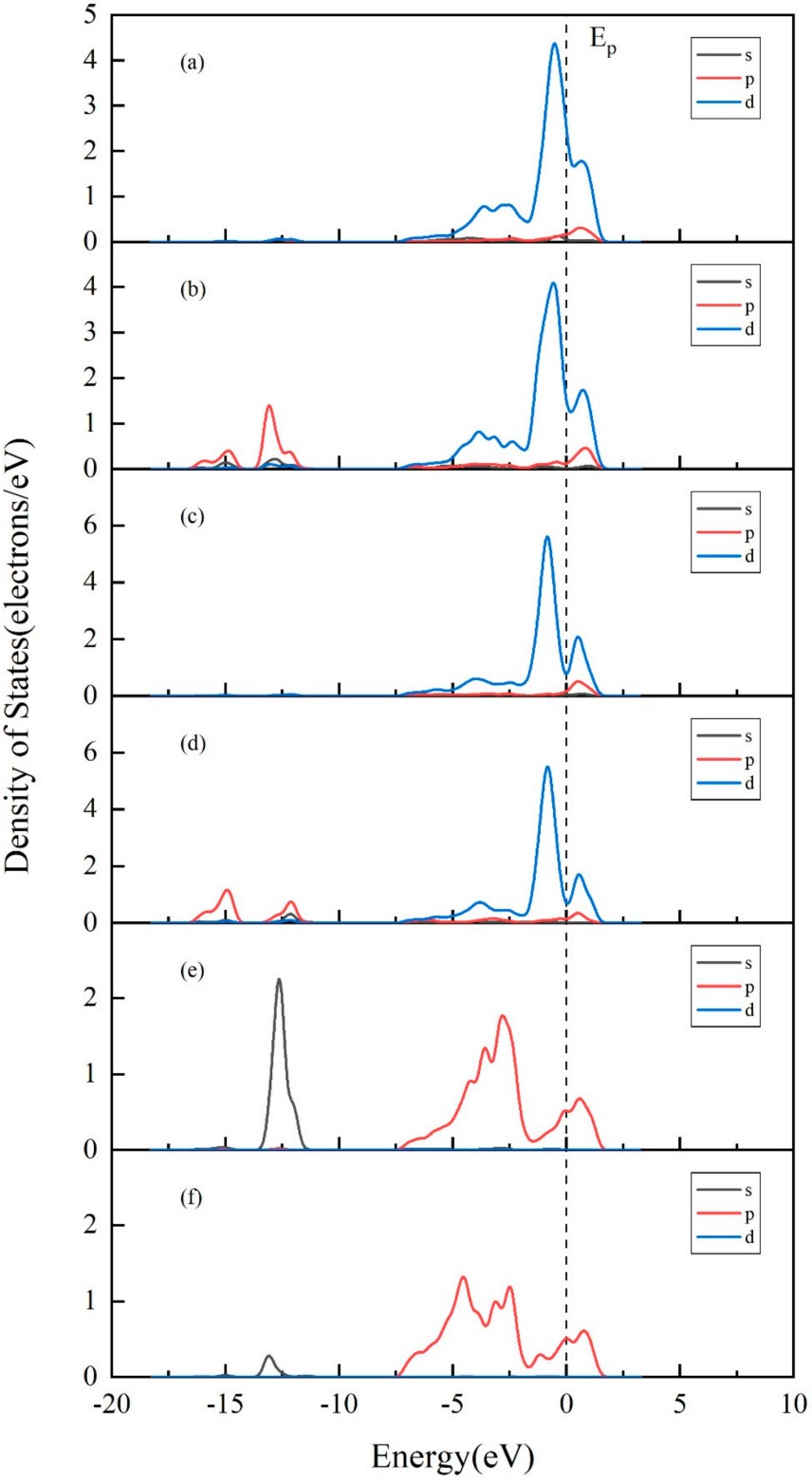

**Figure 5.** The PDOS plots of atoms before and after Au adsorption on pyrite (210) surface: (**a**,**b**) are PDOS before and after adsorption of Fe1 atoms on the surface of pyrite (210); (**c**,**d**) are PDOS before and after adsorption of Fe2 atoms on the surface of pyrite (210); (**e**,**f**) are PDOS before and after adsorption of S2 atoms on the surface of pyrite (210).

After the interaction of Au atoms with Fe atoms on the surface of (210), the nonlocal nature of the 5d orbital is enhanced and the overall density of states develops in the direction of energy reduction. Taking the Fermi energy level as the dividing line, between the left side of the Fermi energy level, $-5$–0 eV is the bonding interaction between Au atomic orbitals and Fe atomic orbitals, and the right side of the Fermi energy level 0–2.5 eV is the anti-bonding interaction between Au atomic orbitals and Fe atomic orbitals. According to the analysis of the peak height of the density of states, the bonding effect is much stronger than the anti-bonding effect, so Au-Fe is bonded as a whole, while the p-orbital peak of Fe atoms increases at lower energy levels, which corresponds to the Mulliken bonding charge change, and the adsorption of Fe1 atoms is stronger than that of Fe2 atoms. The anti-bonding interaction between Au atom orbital and surface S2 atom orbital is weak, bonding is stronger, S2 atom low energy level s orbital peak decreases and reaches a steady state, bonding with gold atoms, so Au atoms can adsorb on the surface of pyrite (210). In addition, Au/Fe is close to the Fermi energy level, the density of states peak is higher, and the adsorption effect of Fe atoms on Au atoms is greater than that of S2 atoms.

## 4. Conclusions

In this work, the microscopic mechanism of gold adsorption on the surface of pyrite was investigated. Different surface models of pyrite were selected, and the adsorption process of gold atoms on pyrite surface was studied by the DFT method. By calculating the surface energy of different surfaces, it is concluded that the pyrite (100) surface is the most stable surface that exists in the thermodynamic state; in addition, the calculation of adsorption energy shows that the (210) surface is the most susceptible to adsorption and has the most stable system structure. Using Mulliken analysis, the adsorption of gold atoms on the pyrite (210) surface was investigated to occur mainly through the combination with surface sulfur and iron atoms. In the adsorption process, sulfur atoms mainly lose electrons and metal atoms mainly gain electrons. Before and after adsorption, PDOS calculations show that the bonding effect of sulfur and iron atoms near the gold atoms is stronger, and the charge transfer of atoms in other parts also occurs on the surface, which is favorable to the adsorption of gold atoms, and the adsorption capacity of iron atoms to gold atoms is greater than that of sulfur atoms. Therefore, in the process of mineralization, gold can be adsorbed on the surface of pyrite, which is an associated mineral of gold. In the high temperature hydrothermal reaction, pH and other impurity ions can also affect the gold adsorption, which needs further study.

**Author Contributions:** C.L. and Y.L. conceived the structure of manuscript and designed the calculation models; Q.C. and Y.Z. provided feasible suggestions; Y.Z. contributed the computation opinions; C.L. performed all the calculations and prepared the original draft; C.L. and Y.L. wrote the manuscript. All authors have read and agreed to the published version of the manuscript.

**Funding:** This research was supported financially by the Key Project of National Natural Science Foundation of China (No. 41530315).

**Institutional Review Board Statement:** Not applicable.

**Informed Consent Statement:** Not applicable.

**Data Availability Statement:** Not applicable.

**Conflicts of Interest:** The authors declare no conflict of interest.

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
