# Peer review of "Atomic Model of Gold Adsorption onto the Pyrite Surface with DFT Study"

_minerals, doi:10.3390/min12030387_

Round 1
Reviewer 1 Report
The present manuscript describes, the adsorption mechanism of gold on the surface of pyrite using DFT. The results are interesting, but not adequate to get published in the present format. I recommend this for publication after addressing the following comment.
1. How authors are sure about the adsorbed gold atom position on the (210), (111) and (100) surfaces is most stable one. For. e.g. in the case of (100) several positions like top, bridge and hollow are available for gold atom adsorption. If authors has done such study, the results need to be supplemented here.
2. In the Table.3, the charge on Au atom is positive (0.03) in the case of (111) surface. Whereas, it is negative for other two surfaces. The reason for this may be due to adsorption position of Au atom on (111) surface. Please consider the other position of Au atom on (111) surface.
3. Authors are requested to study Au adsorption on at least 2-3 sites on each of the surface {(210), (111) and (100) surfaces}. This will improve the quality of the manuscript.
4. In the Table 4. title and contents are confusing(atoms before and after Au adsorption on pyrite (100) surface; whereas content has only one value).
Minor change
1. Please indicate S 1, S 2, S 3, Fe 1..etc. in Figure.2. In the present format it is difficult to corroborate the results in Table 3.

Reviewer 2 Report
The manuscript entitled "Atomic Model of Gold Adsorption onto the Pyrite Surface with DFT Study" investigated the adsorption mechanism of gold on pyrite surfaces by considering different surfaces (100, 111, and 210). The organization of paper is very good and the usage of English is fine. In order to show the novelty and necessity of such a research in this field, the authors not only present adequate number of studies in introduction section but also discussed their results within each other and with the ones reported in literature. I think the paper can be acceptable and publishable in its current form. But only one point took my attention and if possible, it will be better if some explanatory statements can be added to In section 2.2, the effect of surface energy was calculated for pyrites of different surfaces, and a gradual increase was shown as a function of atomic layers (Table 3), so can you please add an additional statement to this section or in future sections for explaining the reason of this variation?
Overall, my suggestion is the acceptance and publication of this paper in its current form.
Author Response
Thank you for your affirmation and hard work on our manuscript. Your suggestions and comments on our manuscript are very valuable and helpful. In addition, according to the suggestions and comments of the other two reviewers, the manuscript has been revised, hoping to meet the requirements of journal publication.
Reviewer 3 Report
See the attached file.

Round 2
Reviewer 1 Report
I thank the author for replying most of the queries.
The authors have revised the manuscript with their best ability according to the comments raised. All the revisions and corrections are acceptable. So, my recommendation is acceptance for publication.
Author Response

(The authors gave the same response as above.)

Reviewer 3 Report
- I still feel that the connection to improvement of the macroscopic minerallization mechanism is missing. You added in the main text half a sentence "which has certain theoretical and practical application reference values for improving the flotation technology of pyrite and enriching the basic theory of flotation.", and in the response a bit more elaborate discussion. One expects to see more changes in the main text elucidating these aspects, and not only in the response.
- Although, it is good that you added atom labels to the figure 2, I was talking about Tables 3 and 4 that are not easily understood and need to be made more clear. This is still missing.
- "Using the DFT method, it helps to understand the adsorption mechanism of gold nanoparticles by constructing models of individual gold atoms adsorbed on different surfaces of pyrite, so this study focuses on the calculation of gold with 0 valence on the surface of pyrite" - how exactly information of a single atom adsorption is translated to a nanoparticle adsorption? Nanoparticles have surfaces as well, therefore their mechanism of adsorption can not be simply extrapolated from a single atom.
- "Your suggestion is very reasonable, the adsorption model of this study is higher than the pyrite surface, slightly less than the vacuum layer thickness of the position of the added gold atoms, after geometric optimization, to obtain a stable model of the pyrite surface, did not consider the adsorption of different positions, this problem, we will do further exploration in the next study." - I don't understand this response at all. The only thing I understood is that you didn't try other initial positions.
Round 3
Reviewer 3 Report
I still find Table 3 not satisfactory. I see that the table is based on table from previous paper, but this table the authors refer to is well made and easy to read. The current Table 3 is confusing.
Author Response
Please see the attachment.

This manuscript is a resubmission of an earlier submission. The following is a list of the peer review reports and author responses from that submission.